# The Cell Adhesion Molecule L1 Interacts with Methyl CpG Binding Protein 2 via Its Intracellular Domain

**DOI:** 10.3390/ijms23073554

**Published:** 2022-03-24

**Authors:** Gabriele Loers, Ralf Kleene, Maria Girbes Minguez, Melitta Schachner

**Affiliations:** 1Zentrum für Molekulare Neurobiologie, Universitätsklinikum Hamburg-Eppendorf, Martinistr. 52, 20246 Hamburg, Germany; gabriele.loers@zmnh.uni-hamburg.de (G.L.); ralf.kleene@zmnh.uni-hamburg.de (R.K.); magirmin@gmail.com (M.G.M.); 2Keck Center for Collaborative Neuroscience, Department of Cell Biology and Neuroscience, Rutgers University, 604 Allison Road, Piscataway, NJ 08854, USA

**Keywords:** cell adhesion molecule L1, proteolytic processing, nuclear import, methyl-CpG-binding protein 2

## Abstract

Cell adhesion molecule L1 regulates multiple cell functions, and L1 deficiency is linked to several neural diseases. Recently, we have identified methyl CpG binding protein 2 (MeCP2) as a potential binding partner of the intracellular L1 domain. By ELISA we show here that L1’s intracellular domain binds directly to MeCP2 via the sequence motif KDET. Proximity ligation assay with cultured cerebellar and cortical neurons suggests a close association between L1 and MeCP2 in nuclei of neurons. Immunoprecipitation using MeCP2 antibodies and nuclear mouse brain extracts indicates that MeCP2 interacts with an L1 fragment of ~55 kDa (L1−55). Proximity ligation assay indicates that metalloproteases, β-site of amyloid precursor protein cleaving enzyme (BACE1) and ɣ-secretase, are involved in the generation of L1−55. Reduction in MeCP2 expression by siRNA decreases L1-dependent neurite outgrowth from cultured cortical neurons as well as the migration of L1-expressing HEK293 cells. Moreover, L1 siRNA, MeCP2 siRNA, or a cell-penetrating KDET-containing L1 peptide leads to reduced levels of myocyte enhancer factor 2C (Mef2c) mRNA and protein in cortical neurons, suggesting that the MeCP2/L1 interaction regulates Mef2c expression. Altogether, the present findings indicate that the interaction of the novel fragment L1−55 with MeCP2 affects L1-dependent functions, such as neurite outgrowth and neuronal migration.

## 1. Introduction

The cell adhesion molecule L1 not only plays crucial roles during the development of the nervous system, but it also regulates important functions in the adult nervous system under physiological and pathological conditions (for reviews and references see, [1,2,3]). In humans, L1 is associated with neurological and psychiatric disorders, such as fetal alcohol syndrome, Hirschsprung’s disease, schizophrenia, Alzheimer’s disease, and autism [4,5,6,7,8,9,10]. Mutations in the L1 gene located on the X chromosome lead to mild to severe congenital developmental disorders in humans (altogether called L1 syndrome), which are characterized, among other features, by hydrocephalus, mental retardation, spastic paraplegia, hypoplasia of the corpus callosum, a shuffling gait, and adducted thumbs [11]. Numerous studies in mice have shown that L1 acts predominantly through homophilic but also heterophilic interactions, and that the beneficial functions of L1 depend on its proteolytic cleavage (for reference see, for instance, [12]).

Important physiological processes such as neuronal differentiation, migration and survival, axon outgrowth and fasciculation, synaptogenesis, myelination, synaptic plasticity, learning and memory, as well as regeneration after injury are all abnormal in the absence of L1, leading to morphological and functional changes as well as behavioral disturbances [3,13,14,15,16,17,18,19,20,21,22].

First indications suggesting a functional interdependence of L1 and MeCP2 were gained by studying reprogrammed human stem cells from a Rett syndrome individual: reduced L1 expression as well as reduced neurite outgrowth and neuronal cell survival were observed, and expression of wild-type MeCP2 in these cells resulted in increased L1 levels, neurite outgrowth, and neuronal cell survival [9]. Moreover, deficiency in neurite outgrowth could be normalized by transfection of these cells with full-length L1 [9].

Following up on these observations, we have recently found that MeCP2 in a nuclear mouse brain extract associates with immobilized recombinant intracellular L1 domain (L1-ICD) [23]. This relationship is deemed interesting, since MeCP2 is a transcription factor that is active in the cell nucleus, and its activity could be changed by interacting with L1. MeCP2 binds not only to methylated CpG islands but also to unmethylated DNA [24]. Especially in nerve cells, MeCP2 plays an important role in the repression and activation of the transcription of many genes [25,26]. Mutations in the MeCP2 gene, which, like L1, is located on the X chromosome in the Xq28 chromosomal region, cause Rett syndrome, a developmental disorder that leads to prenatal or perinatal death of boys and severely affects girls [27,28]. Individuals with Rett syndrome lose many early childhood skills. As adults, they typically exhibit symptoms of autism, disorders of motion coordination with spasms primarily of legs, epileptic seizures, frequent ataxia, and hand movement stereotypes, as well as other symptoms in social interactions. Some individuals are severely impaired in mental activities and either do not develop language skills or tend to lose learned language. Morphological correlates are frequent hypoplasia of the corpus callosum and many other substructural abnormalities [29,30].

Among the MeCP2-deficient mouse lines investigated so far, several show a similar abnormal phenotype, the most remarkable of which are a staggering gait, other disturbances of motor coordination and hypoactivity, as well as changes in anxiety, learning capabilities, and social behavior [31,32]. Morphological abnormalities, changes in gene expression, and impaired neuronal survival have been observed in MeCP2-deficient and MeCP2-overexpressing mice during postnatal development [33,34,35,36,37,38].

Since mutations in L1, absences of wild-type L1 or wild-type MeCP2, as well as increased expression of MeCP2 by duplication of the MeCP2 gene lead to similar phenotypes, it is deemed likely that a balanced interaction of L1 with MeCP2 is essential for normal development of the nervous system.

In the present study, we have identified an L1 fragment of approximately 55 kDa (L1−55) comprising the intracellular domain as a novel binding partner of MeCP2. We also present evidence that this interaction occurs in the cytoplasm and nucleus of cultured cerebellar and cortical neurons and that MeCP2 contributes to regulating L1-dependent functions, such as neurite outgrowth and neuronal migration.

## 2. Results

### 2.1. A Nuclear L1 Fragment of ~55 kDa Binds to MeCP2 via Its Intracellular Domain

Because of the functional importance of MeCP2 and L1, we further investigated their interactions. As a first step, we performed immunoprecipitation with L1 and MeCP2 antibodies using nuclear brain fractions from wild-type and L1-deficient mice. By Western blot analysis, a ~80 kDa MeCP2-immunopositive band was detected in the L1 immunoprecipitate from wild-type mice, whereas this band was not detectable in the L1 immunoprecipitate from L1-deficient mice and in control immunoprecipitates with a non-immune mouse antibody (Figure 1a). Western blot analysis with the antibody C-2 directed against an epitope in L1-ICD showed a predominant band of ~55 kDa and a minor band of ~40 kDa in the MeCP2 immunoprecipitates, but not in the control immunoprecipitates and in the nuclear extract (Figure 1b). Using a differently prepared nuclear extract, the ~55 kDa L1 fragment was observed in the MeCP2 immunoprecipitate and in the nuclear extract (Figure 1c), but not in the control immunoprecipitates. The ~40 kDa band was only observed in the MeCP2 immunoprecipitate, but not in the nuclear extract and control immunoprecipitates (Figure 1c), suggesting that the ~40 kDa band is an unspecific contamination. These results indicate that a 55 kDa L1 fragment containing the intracellular domain associates with MeCP2.

By ELISA, a concentration-dependent and saturable binding of recombinant L1-ICD to substrate-coated recombinant MeCP2 was observed, whereas the recombinant intracellular domain of the close homolog of L1 (CHL1-ICD) did not bind to MeCP2 (Figure 1d). A concentration-dependent binding of L1-ICD, but not CHL1-ICD, was also observed in a label-free binding assay (Figure 1e). These results show that L1-ICD directly binds to MeCP2.

To determine the site(s) in L1-ICD that bind(s) to MeCP2, five peptides which cover the total L1-ICD sequence were assayed in a competition ELISA for their capacity to inhibit the binding of L1-ICD to MeCP2. Peptides comprising L1-ICD amino acids 1–35 (P1), 23–57 (P2), 14–29 (Pb), 25–43 (Pc), and 19–32 (Ps) reduced the binding of L1-ICD to MeCP2, whereas peptides comprising amino acids 1–17 (Pa), 55–73 (P3), and 70–89 (P4) did not affect the binding (Figure 1f). The peptide comprising amino acids 87–114 (P5) did not reduce the binding, but showed a yet unexplained increase in binding as measured by signal intensity (Figure 1f). Since all peptides that disturb the binding of L1-ICD to MeCP2 contain the sequence KDET, we concluded that the KDET sequence mediates the binding of L1-ICD to MeCP2. Of note, the KDET motif is present not only in the murine and human L1-ICD sequence, but also in the L1-ICD sequence of many, if not all, mammalian, fish, avian, and amphibian species.

### 2.2. MeCP2 Co-Localizes with L1 in the Cytoplasm and Nucleus of Cerebellar and Cortical Neurons

To analyze whether MeCP2 and L1 interact also in a cellular context, proximity ligation assay was performed with cultured wild-type and L1-deficient cerebellar neurons. Using a mouse L1-ICD antibody and a rabbit MeCP2 antibody, positive signals were observed as red puncta in the cytoplasm and nuclei of wild-type cerebellar neurons but not in L1-deficient neurons (Figure 2a), indicating a close association and a specific interaction of MeCP2 and L1 in the cytoplasm and nucleus. After treatment of cerebellar neurons with the function-triggering L1 antibody 557, which increases the levels of L1 fragments, the average numbers of MeCP2/L1-positive red puncta per cell were approximately two times higher in comparison to neurons without treatment (Figure 2b,c). These results indicate that increased L1 levels lead to an enhanced interaction of MeCP2 with L1.

To further support the notion that L1 interacts with MeCP2 in the nucleus, images were taken from z-axis stacks. In several cells, most z-axis stacks showed only red puncta, whereas several stacks displayed purple puncta due to the overlap between the blue nuclear DAPI stain and the red signal of the proximity ligation assay, which are indicative of a nuclear interaction of L1 and MeCP2 (Figure 2d), supporting the notion that L1 interacts with MeCP2 in the nucleus.

MeCP2/L1-positive puncta were also observed in cortical neurons (Figure 2e). However, the numbers of MeCP2/L1-positive puncta were not enhanced by treatment with function-triggering L1 antibody 557 (Figure 2e,f). Images of z-axis stacks displayed purple puncta in the nucleus of a cortical neuron (Figure 2g), indicating that L1 and MeCP2 interact in the nuclei of cortical neurons.

### 2.3. L1−55 Is Generated by γ-Secretase Cleavage

Recently, we identified a ~55 kDa L1 fragment (L1−55) interacting with heterochromatin protein 1 isoforms α, β, and γ via the KDET motif, and we proposed that the generation of this fragment involves the metalloprotease ADAM-10 and/or -17, BACE1, and γ-secretase [39]. Since it was very likely that HP1-binding L1−55 is identical to the ~55 kDa L1 fragment that binds to MeCP2, we analyzed whether the generation of the MeCP2-binding ~55 kDa L1 fragment also involves ADAM-10 and/or -17, BACE1, and γ-secretase. To this aim, cultured cortical neurons were treated with metalloprotease inhibitor GM6001, BACE1 inhibitor LY2886721, or γ-secretase inhibitor DAPT and were subjected to proximity ligation assay. In comparison to the vehicle-treated neurons, neurons treated with DAPT, GM6001, and LY2886721 showed a strong reduction in numbers of L1/MeCP2-positive puncta per cell, whereas numbers were not altered in neurons treated with the serine protease inhibitor aprotinin (Figure 3a). In cultured cerebellar neurons, DAPT, GM6001, and LY2886721 also resulted in a strong reduction in numbers of L1/MeCP2-positive puncta per cell (Figure 3b). This result indicates that the generation of the MeCP2-associated ~55 kDa L1 fragment involves cleavage by ADAM-10 and/or -17, BACE1, and γ-secretase, as observed for the HP1-associated L1−55 fragment, suggesting that the MeCP2-associated L1 fragment represents L1−55.

L1−55 was detectable in MeCP2 immunoprecipitates but not in control immunoprecipitates from nuclear extracts from cortical neurons (Figure 3c). Levels of L1−55 were decreased in MeCP2 immunoprecipitates from nuclear extracts of DAPT-treated cortical and cerebellar neurons (Figure 3c,d). When compared to the L1−55 levels in MeCP2 immunoprecipitates from nuclear extracts of untreated neurons, L1−55 levels were not altered in immunoprecipitates from nuclear extracts of L1 antibody 557-treated cortical neurons (Figure 3c), whereas the levels were increased in MeCP2 immunoprecipitates from nuclear extracts of L1 antibody 557-stimulated cerebellar neurons (Figure 3d). In contrast to the major L1−70 fragment, L1−55 was often hardly detectable in nuclear extracts from cortical and cerebellar neurons (Figure 3c,d), most likely due to low concentrations of this fragment in cultured neurons. A ~40 kDa band was only occasionally seen in MeCP2 immunoprecipitates and nuclear extracts (Figure 3c,d), suggesting that this band is unspecific. In summary, the results indicate that inhibition of γ-secretase activity reduces the levels of MeCP2 bound to L1−55.

### 2.4. Reduction of MeCP2 Expression Inhibits L1-Dependent Migration of HEK293 Cells

To investigate whether the interaction between L1 and MeCP2 plays a role in L1-mediated cell migration, HEK293 cells, which do not express L1, were mock-transfected or transfected with MeCP2 siRNA alone or together with an L1-encoding plasmid. These cells were subjected to a scratch assay that allows determining cell migration in vitro. The extent of gap closure was calculated and used as a measure of cell migration. In comparison to mock-transfected cells, the migration of L1-expressing cells was increased (Figure 4a,b). The migration of mock-transfected cells and of cells transfected with MeCP2 and control siRNA alone was similar, indicating that the reduction of MeCP2 expression in the absence of L1 expression did not affect cell migration (Figure 4b). The migration of L1-expressing cells was reduced after transfection with MeCP2 siRNA but was unaltered after transfection with control siRNA (Figure 4a,b). Western blot analysis of cells transfected with the control or with MeCP2 siRNA and L1 plasmid showed reduced MeCP2 levels (18.7 ± 2.8%; *n* = 3; *p* < 0.001, one-way ANOVA with Bonferroni’s multiple comparison test) in L1-expressing cells after transfection with MeCP2 siRNA relative to L1-expressing cells transfected with control siRNA (Figure 4c), confirming the reduction of expression of MeCP2 in L1-expressing cells by the corresponding siRNA. In summary, the results indicate that L1-mediated cell migration depends on MeCP2.

### 2.5. Reduction of MeCP2 Expression Inhibits L1-Dependent Neurite Outgrowth from Cortical Neurons

Next, we investigated whether the interaction of L1 with MeCP2 plays a role in L1-mediated neurite outgrowth. Since cerebellar neurons are not suitable for transfection with siRNAs and plasmids, we used cortical neurons for transfection. Compared to mock-transfected neurons, transfection of neurons with MeCP2 siRNA led to reduced neurite outgrowth, whereas transfection with control siRNA had no effect on neurite outgrowth (Figure 5a). Western blot analysis showed reduced MeCP2 levels (42.4 ± 8.5%; *n* = 3; *p* < 0.01 one-way ANOVA with Tukey’s multiple comparison test) in neurons after transfection with MeCP2 siRNA relative to the levels in neurons transfected with control siRNA (Figure 5b), confirming the reduction of MeCP2 expression in cortical neurons by the corresponding siRNAs.

We also determined by proximity ligation the number of L1/MeCP2-positive puncta in cortical neurons after transfection with L1 siRNA or MeCP2 siRNA and found that the numbers were reduced by more than 70% (Figure 5c,d), indicating that the L1/MeCP2 interactions are strongly reduced by siRNAs. Of note, the reduction was similar in nuclear and non-nuclear locations. Since treatment with the function-triggering L1 antibody 557 increases neurite outgrowth [40,41,42], we treated cortical neurons with this antibody to induce L1-dependent neurite outgrowth and to determine whether MeCP2 is involved in this L1-induced enhancement of neurite outgrowth. In addition, we analyzed whether enhancement of L1 expression by transfection with L1 plasmid or reduction of L1 expression by transfection with L1 siRNA would affect neurite outgrowth. Treatment with L1 antibody 557 promoted neurite outgrowth from mock-transfected neurons (Figure 5a). This enhanced neurite outgrowth was slightly increased after transfection with the L1-encoding plasmid but was prevented by transfection with L1 siRNA (Figure 5a). Transfection with control siRNA neither affected the promotion of neurite outgrowth by L1 antibody 557 treatment and by L1 plasmid-mediated L1 overexpression, nor did it affect the diminished neurite outgrowth from neurons co-transfected with L1 siRNA (Figure 5a). However, L1 antibody 557-triggered and L1 plasmid-mediated enhancement of neurite outgrowth was reduced by transfection with MeCP2 siRNA, whereas MeCP2 siRNA did not further decrease neurite outgrowth in neurons co-transfected with L1 siRNA (Figure 5a). In summary, these results indicate that L1-mediated neurite outgrowth depends on the interaction of L1 with MeCP2.

### 2.6. Gene Expression of Mef2c in Cortical Neurons Is Reduced by MeCP2 siRNA, L1 siRNA, and the KDET Motif-Carrying L1 Peptide

To investigate whether the L1/MeCP2 interaction affects gene expression of the selected genes, we used L1 siRNA and MeCP2 siRNAs to reduce either L1 or MeCP2 levels, thereby reducing L1/MeCP2 interactions in cortical neurons. As possible targets we chose the brain-derived neurotrophic factor (BDNF), Wnt family member 7A (Wnt7a), and Hes family BHLH transcription factor 5 (Hes5), since comparisons of transcriptome data from the cerebellum of L1-deficient mice [43] and MeCP2-deficient mice [44] indicate that these genes are downregulated in MeCP2- and L1-deficient mice. In addition, we analyzed two other possible targets: myocyte enhancer factor 2C (Mef2c) and ataxin 2 binding protein 1 (A2bp1), the expression of which is repressed in mice that lack MeCP2 and is enhanced in mice overexpressing MeCP2 [45]. The expression of BDNF, Hes5, and Wnt7a was neither affected by L1 siRNA nor by MeCP2 siRNA, whereas Mef2c mRNA levels were reduced by L1 siRNA and MeCP2 siRNAs (Figure 6a). The reduction of A2bp1 gene expression by L1 siRNA (0.71 ± 0.05) or by MeCP2 siRNA (0.76 ± 0.06) did not reach the level of significance.

Interestingly, after the treatment of neurons with L1 antibody 557, Mef2c mRNA levels were not altered when compared to mock-treated neurons, but they were reduced by L1 siRNA and MeCP2 siRNAs to a similar extent, as seen in siRNA-transfected unstimulated neurons (Figure 6a).

The reduction of Mef2c mRNA levels by MeCP2 and L1 siRNAs coincided with reduced Mef2c protein levels (Figure 6b,c). These results suggest that the reduction of L1/MeCP2 interaction by the siRNA-mediated decrease in L1 or MeCP2 levels leads to repression of Mef2c mRNA and protein expression.

To further support this notion, we used the cell-penetrating tat-L1/KDET peptide to disturb the KDET motif-mediated interaction of L1 with MeCP2 and analyzed the effect of this peptide on Mef2c gene expression. In the presence of the peptide, the Mef2c mRNA levels were reduced in non-stimulated and L1 antibody 557-stimulated cortical neurons (Figure 6a). Similarly, the Mef2c protein levels were reduced by the peptide (Figure 6b,c). The reduced Mef2c mRNA and protein levels coincided with a reduction of L1−55 levels in MeCP2 immunoprecipitates from nuclear extracts of tat-L1/KDET peptide-treated cortical and cerebellar neurons relative to immunoprecipitates from nuclear extracts of untreated cells (Figure 7a,b). Moreover, we found reduced numbers of L1/MeCP2-positive puncta after treatment of non-stimulated and L1 antibody 557-stimulated cortical neurons with the tat-L1/KDET peptide by proximity ligation (Figure 7c). These results indicate that the L1/MeCP2 interactions are reduced by the peptide. Of note, similar numbers of L1/MeCP2-positive puncta were found in non-stimulated and antibody 557-stimulated cortical neurons (Figure 7c), indicating that the interaction between L1 and MeCP2 is not enhanced by the stimulation of cortical neurons with antibody 557. Interestingly, the interaction between L1 and HP1, which is also mediated by the KDET motif [39], was not affected by the peptide in the absence of antibody 557, whereas the antibody 557-induced L1/HP1 interaction was inhibited by the peptide (Figure 7d).

In summary, these results indicate that the L1/MeCP2 interaction is required for regulation of Mef2c expression.

## 3. Discussion

After having identified MeCP2 as a potential binding partner of L1-ICD [23], we verified here the direct binding of MeCP2 to L1-ICD by ELISA. This observation is supported by proximity ligation, which showed a close association of L1 and MeCP2 in cerebellar and cortical neurons. These interactions occur in the cytoplasm near the nucleus and in the nucleus. Co-immunoprecipitation of L1−55 with MeCP2 antibodies suggests that this L1 fragment interacts with MeCP2. Since the reduction of MeCP2 protein expression diminishes the migration of L1-expressing HEK cells and L1-dependent neurite outgrowth from cortical neurons, we propose that the interaction of L1−55 with MeCP2 is required for regulating L1-dependent functions, such as cell migration and neurite outgrowth.

MeCP2 not only binds to DNA sequences containing methylated CpG dinucleotides but also to unmethylated DNA, thereby functioning both as a transcriptional repressor and an activator. Moreover, MeCP2 also contributes to the organization of the chromatin structure and is involved in splicing mechanisms [24,26,46,47,48,49,50,51,52,53,54,55,56]. Mutations in the X-linked MeCP2 gene cause Rett syndrome, which is characterized—similar to L1 syndrome—by severe mental retardation and autism [55,57].

The functional interplay of L1 and MeCP2 and the interaction of L1 and MeCP2 with other interaction partners may affect L1- and MeCP2-mediated cellular functions. It is noteworthy in this respect that MeCP2 interacts with HP1γ and that the MeCP2/HP1γ interaction contributes to the reorganization of heterochromatin during myogenesis [58,59]. Interestingly, HP1α, -β, and -γ, like MeCP2, interact with L1−55 [39] and regulate L1 functions, such as cell migration and neurite outgrowth. Histone H1.4, another nuclear L1 binding partner [23], competes with MeCP2 for common chromatin binding sites and decreases the binding of HP1γ to DNA [60]. Since histone H1 is involved in chromatin folding, CpG methylation, DNA repair, and regulation of the expression of imprinted genes or genes on the X chromosome [61,62,63,64], it is conceivable that the interaction of histone H1 with L1 and/or MeCP2 could modulate these functions of histone H1. The findings that L1 and MeCP2 and their interaction partners HP1 and histone H1 regulate multiple cellular events, suggest that the interplay of L1−55 and MeCP2, HP1, histone H1, or other binding partners regulates not only L1-dependent functions, but also MeCP2-, HP1- or histone H1-related functions during development and in adulthood. It is tempting to speculate that the L1−55/MeCP2 complex modulates the binding of MeCP2 to DNA and may thus alter MeCP2-regulated gene expression. Notably, the reduction of L1 and MeCP2 levels leads to a reduction in Mef2c expression. Mef2c is an important transcription factor during neural development, being essential for early neurogenesis, neuronal migration, and differentiation [65,66]. Mutations of Mef2c are associated with the Mef2c haploinsufficiency syndrome, which is a severe developmental disorder associated with intellectual disabilities, developmental delay, lack of speech, stereotypic movements, limited walking abilities or the inability to walk, seizures, and poor eye contact [65,66,67,68,69,70,71,72]. Mef2c is also linked to Rett syndrome or severe Rett-like encephalopathies as well as schizophrenia and autism spectrum disorders characterized by dysmorphic features and cerebral malformations [73,74,75,76,77,78,79]. It is noteworthy in this context that heterozygous Mef2c mice display autism-related behaviors [74], similar to L1-heterozygous mice [10].

We expect that the interaction of L1−55 with MeCP2 can also affect other MeCP2- and L1-dependent nuclear functions, such as mRNA processing, splicing, chromatin folding, and DNA repair. The L1−55/MeCP2 interaction might also affect the interplay of other L1 fragments with their nuclear binding partners, such as topoisomerase I [80]. Ablation or disruption of L1 and MeCP2 interactions may aggravate development and brain function in adulthood, resulting in Rett or L1 syndromes, which display similar deficits. Altogether, we believe that the present findings contribute to a deeper understanding of L1 and MeCP2 interactions in the developing and adult nervous system under normal and pathological conditions, for example, in autism.

## 4. Materials and Methods

### 4.1. Animals

L1-deficient mice have been described [16]. Mice were bred and maintained at the Universitätsklinikum Hamburg-Eppendorf at 25 °C on a 12 h light/12 h dark cycle with ad libitum access to food and water. C57BL/6J males and females and L1-deficient males and their wild-type male littermates were used for all experiments. All animal experiments were conducted in accordance with the German and European Community laws on the protection of experimental animals and approved by the local authorities of the State of Hamburg (animal permit numbers N19/004_ZuchtNeuro, ORG 679 Morph and ORG 1022). The manuscript was prepared following the ARRIVE guidelines for animal research [81].

### 4.2. Reagents and Antibodies

Mouse monoclonal L1 antibody C-2 (NCAM-L1; sc-514360; no RRID available) against the intracellular L1 domain was obtained from Santa Cruz Biotechnology (Dallas, TX, USA). Rat monoclonal function-triggering L1 antibody 557 has been described [82]. The rabbit monoclonal MeCP2 antibody D4F3 (Cell Signaling Technology Cat# 3456, RRID:AB_2143849) and the rabbit monoclonal Mef2c antibody D80C1 (Cell Signaling Technology Cat# 5030, RRID:AB_10548759) were obtained from Cell Signaling Technology (Danvers, MA, USA). The goat anti-CHL1 antibody C-18 against an epitope in the intracellular domain (Santa Cruz Biotechnology Cat# sc-34986, RRID:AB_1121563) and the mouse antibody against GAPDH (Santa Cruz Biotechnology Cat# sc-47724, RRID:AB_627678) were obtained from Santa Cruz Biotechnology. Secondary antibodies were obtained from Dianova (Hamburg, Germany). The production and purification of recombinant His-tagged intracellular domains of mouse L1 (L1CAM_MOUSE, P11627; aa 1147–1260) (L1-ICD) and mouse CHL1 (NCHL1_MOUSE, P70232; aa 1105–1209) (CHL1-ICD) have been described [83,84,85].

Recombinant full-length human MeCP2 (amino acids 1–486) with GST-tag (#H00004204-P01) was obtained from Abnova (Taipei, Taiwan). Recombinant full-length human MeCP2 with a Flag-tag was obtained from Genway (Palo Alto, CA, USA). Synthetic L1 peptides P1 (CFIKRSKGGKYSVKDKEDTQVDSEARPMKDETFGE), P2 (RPMKDETFGEYRSLESDNEEKAFGSSQPSLNGDIK), P3 (GDIKPLGSDDSLADYGGSVD), P4 (CSVDVQFNEDGSFIGQYSGK), P5 (CSGKKEKEAAGGNDSSGATSPINPAVALE), Pa (FIKRSKGGKYSVKDKEDTQ), Pb (EDTQVDSEARPMKDET), Pc (KDETFGEYRSLESDNEEK), Ps (DSEARPMKDETFGE), and tat-L1/KDET (YGRKKRRQRRRDSEARPMKDETFGE) comprising the cell-penetrating HIV tat sequence YGRKKRRQRRR [86] and the L1-ICD sequence DSEARPMKDETFGE were obtained from Schafer-N (Copenhagen, Denmark). Mouse MeCP2 siRNA (sc-35893), mouse L1 siRNA (NCAM-L1; sc-43173), and control siRNA (Control siRNA-A; sc-37007) were obtained from Santa Cruz Biotechnology. FuGENE^®^ HD Transfection Reagent (Cat #E2311) was obtained from Promega (Walldorf, Germany).

N-[3-[(4aS,7aS)-2-amino-4a,5-dihydro-4H-furo[3,4-d][1,3]thiazin-7a(7H)-yl]-4-fluorophenyl]-5-fluoro-2-pyridinecarboxamide (LY2886721; CAS 1262036-50-9) (Cay21599-1), (2S)-N-[(3,5-difluorophenyl)acetyl]-L-alanyl-2-phenyl]glycine 1,1-dimethylethyl ester (DAPT; CAS 208255-80-5) (Cay13197-5), (2R)-N4-hydroxy-N1-[(1S)-1-(1H-indol-3-ylmethyl)-2-(methylamino)-2-oxoethyl]-2-(2-methylpropyl)-butanediamide (GM 6001; CAS 142880-36-2) (Cay14533-1), and aprotinin (CAS 9087-70-1) (Cay14716-10) were obtained from Biomol (Hamburg, Germany). 4′,6-diamidino-2-phenylindole (DAPI) was obtained from Thermo Fisher Scientific (Darmstadt, Germany).

### 4.3. Preparation of a Nuclear Protein Extract

Protocol A: Brains of adult mice were mechanically disrupted in a homogenization buffer (0.35 M sucrose, 20 mM Tris/HCl, pH 7.4, 1× protease inhibitor cocktail tablet (Sigma-Aldrich) per 50 mL; 3 mL per brain) by using a Potter-Elvehjem tissue homogenizer with a Teflon pestle (Wheaton potter; VWR International GmbH, Darmstadt, Germany) and by applying 20–25 strokes. This and subsequent steps were carried out at 4 °C. A total of 15 min after homogenization, NP-40 was added to a final concentration of 0.2%, and the samples were vortexed at the highest speed for 10 s and centrifuged at 3000× *g* for 10 min. The pellet was overlaid with 20% iodixanol (OptiPrep™ (60% iodixanol); PROGEN Biotechnik, Heidelberg, Germany) in 20 mM Tris/HCl, pH 7.4, homogenized by applying 10 strokes and centrifuged at 10,000× *g* for 40 min. The pellet was resuspended in homogenization buffer and centrifuged at 3000× *g* for 5 min. This pellet was resuspended in 200 µL Nuclear Extraction Buffer (NEB from Subcellular Protein Fractionation Kit for Tissues; Thermo Fisher Scientific), vortexed at the highest speed for 15 s, incubated for 30 min, and centrifuged at 5000× *g* for 5 min. The supernatant was used as a nuclear protein extract.

Protocol B: Brains of juvenile mice (younger than 20 days old) were homogenized in 2 mL (per brain) ice-cold homogenization buffer (0.32 M sucrose, 20 mM Tris-HCl, 2 mM MgCl_2_, 2 mM CaCl_2_, 1 mM phenylmethylsulfonyl fluoride (PMSF), pH 7.4) per brain using a Potter-Elvehjem. The homogenate was centrifuged at 200× *g* for 15 min at 4 °C. The pellet containing the nuclei was gently resuspended in 5 volumes lysis buffer (homogenization buffer containing 0.1% NP40) and loaded onto a discontinuous sucrose gradient, which consisted of 0.8, 1.0, and 1.2 M sucrose in isotonic buffer (20 mM Tris-HCl, 2 mM MgCl_2_, 2 mM CaCl_2_, 1 mM PMSF pH 7.4). After centrifugation for 2 h at 82,705× *g* at 4 °C, the pellet was resuspended in 1.5 mL homogenization buffer and centrifuged for 20 min at 1000× *g* and 4 °C. The pellet was resuspended in 25 µL Roeder C buffer (20 mM Tris-HCl, 420 mM NaCl, 2 mM MgCl_2_, 2 mM CaCl_2_, 0.5 mM DTE, 1 mM PMSF, 5% glycerol, pH 7.5) per brain and incubated for 30 min at 4 °C, and it was vortexed and homogenized by pipetting up and down. The homogenate was then centrifuged at 100,000× *g* at 4 °C for 30 min, and the supernatant was collected as nuclear extract.

For the preparation of nuclear extracts from cultured neurons, a Subcellular Protein Fractionation Kit for Cells (Thermo Fisher Scientific) was used. Cells were harvested in a Cytoplasmic Extraction Buffer (CEB) using a cell scraper. The NEB fraction was used as nuclear extract for immunoprecipitation.

### 4.4. Co-Immunoprecipitation Using the Nuclear Protein Extract

For co-immunoprecipitation, 200–500 µg of nuclear proteins was incubated with 5 µg of L1 antibody C-2, MeCP2 antibody, or non-immune anti-mouse or anti-rabbit antibodies. The incubation volume was adjusted to 250 µL using phosphate-buffered saline containing Ca^2+^ and Mg^2+^ (PBS), and the samples were incubated overnight at 4 °C with gentle rotation. After the addition of 20 µL Protein G-coupled magnetic bead suspension and incubation of the samples for 2 h at room temperature with gentle rotation, the beads were isolated using a magnet and washed four times with PBS. The beads were suspended in 15 µL SDS sample buffer, boiled for 5 min at 95 °C, and separated using a magnet. The resulting supernatant was subjected to Western blot analysis.

### 4.5. ELISA and Label-Free Binding Assay

For ELISA, 25 µL of 5 µg/mL recombinant GST-tagged MeCP2 was incubated overnight at 4 °C in 384-well microtiter plates with a high binding surface (Corning, Tewksbury, MA, USA). All of the following steps were performed at room temperature. Wells were washed with PBS, treated with blocking solution (1% essentially fatty acid-free bovine serum albumin in PBS) for 2 h, washed again with PBS containing 0.005% Tween 20 (PBST), and incubated with increasing concentrations of recombinant His-tagged intracellular L1 domain (L1-ICD) or CHL1 domain (CHL1-ICD) as ligands for 1 h under gentle agitation. For competition ELISA, 2.5 µM L1-ICD were preincubated for 1 h without or with a 5-fold molar excess of L1 peptides. The mixtures were then incubated with substrate-coated recombinant Flag-tagged MeCP2 (25 µL of 10 µg/mL was used for coating). After washing three times with PBST, an L1 antibody C-2 (1:200) or a CHL1 antibody C-18 (1:200) in blocking solution was applied for 1 h followed by three washes with PBST and incubation with horseradish peroxidase-coupled anti-mouse or anti-goat secondary antibodies (diluted 1:2000 in blocking solution) for 1 h. Wells were washed again with PBST, and 1 mg/mL ortho-phenylenediamine dihydrochloride (Thermo Fisher Scientific) was used for the detection of bound L1-ICD and CHL1-ICD. The reaction was terminated by the addition of 25 µL 2.5 M sulphuric acid. Absorbance was measured at 492 nm with an ELISA reader (µQuant; BioTek, Bad Friedrichshall, Germany).

In label-free binding assays, L1-ICD and CHL1-ICD were used as soluble binding partners, and a 384-well plate with a TiO_2_ biosensor surface (SRU Biosystems, Woburn, MA, USA) was used for substrate coating. Binding was determined by measuring the peak wavelength shift of reflected light using the BIND Technology (SRU Biosystems) as described [87].

### 4.6. Transfection of HEK293 Cells and Scratch Wound Assay

HEK293TN cells (BioCat, Heidelberg, Germany) were plated in 6-well plates (Greiner Bio-One, Frickenhausen, Germany) and maintained at 37 °C in a 5% CO_2_ atmosphere in Dulbecco’s modified Eagle’s medium (DMEM) supplemented with L-glutamine, 4.5 mg/mL glucose, 10% fetal calf serum (DMEM/FCS), and 100 µg/mL penicillin/streptomycin. At a confluency of 50–70%, the medium was replaced with DMEM/FCS without penicillin/streptomycin, and the cells were transfected with 1 µg plasmid containing L1 cDNA, 7 µL 10 µM siRNA, and 3 µL FuGENE transfection reagent in 100 µL DMEM. After 16 h, the medium was replaced with DMEM/FCS containing penicillin/streptomycin. After further incubation for 8 h, the cell layer was subjected to scratch injury using a 50 µL pipette tip. After 24 h, cells were fixed in 4% formaldehyde for 20 min at room temperature. The gap size was measured with an Axiovert microscope with the AxioVision 4.6 imaging system (Carl Zeiss, Oberkochen, Germany).

### 4.7. Cultures of Cerebellar and Cortical Neurons

Cerebellar neurons were prepared from cerebella of 6- to 8-day-old mice. Cerebella were incubated with 10 mg/mL trypsin and 0.5 mg/mL DNase I (Sigma-Aldrich) in Hanks’ balanced salt solution (HBSS) for 15 min at 37 °C, washed with HBSS, mechanically dissociated and centrifuged at 100× *g* for 15 min. Cells were then diluted in Neurobasal A medium (Thermo Fisher Scientific) supplemented with 2 mM L-glutamine (Invitrogen), 4 nM L-thyroxine (Sigma-Aldrich), 1 mg/mL BSA (Sigma-Aldrich), 12.5 μg/mL insulin (Sigma-Aldrich), 30 nM sodium selenite (Sigma-Aldrich), 100 μg/mL transferrin (Merck Biosciences, Darmstadt, Germany), 0.1 mg/mL streptomycin, and 100 U/mL penicillin (Invitrogen). For the proximity ligation assay, cells were seeded onto poly-L-lysine-coated 12 mm coverslips in a 24-well plate at a density of 2.5 × 10^5^ cells per well. For the determination of protein levels, cells were seeded at a density of 2 × 10^6^ cells per well of a 6-well plate coated with poly-L-lysine, and for neurite outgrowth, cells were seeded at a density of 5 × 10^4^ cells per well of a 48-well plate coated with poly-L-lysine (Sigma-Aldrich).

For the culturing of cortical neurons, cerebral cortices were dissected from 15.5- to 16.5-day-old embryos and incubated in 0.025% trypsin (Sigma-Aldrich) in HBSS at 37 °C for 30 min. The cortices were then incubated in HBSS containing 1% BSA (Sigma-Aldrich) and 1% trypsin inhibitor (T-6522, Sigma-Aldrich) at 37 °C for 5 min. After washing in HBSS, the tissue was mechanically dissociated, and the dissociated cells were cultured in Neurobasal medium (Thermo Fisher Scientific) supplemented with 1% B-27 (Thermo Fisher Scientific), 2 mM L-glutamine (Thermo Fisher Scientific), 100 U/mL penicillin (Thermo Fisher Scientific), and 100 μg/mL streptomycin (Thermo Fisher Scientific). For the proximity ligation assay, cells were seeded onto poly-L-lysine-coated 12 mm coverslips in a 24-well plate at a density of 2.5 × 10^5^ cells per well. For the determination of protein expression or neurite outgrowth, cells were seeded at a density of 2 × 10^6^ cells per well of a 6-well plate coated with poly-L-lysine or at a density of 5 × 10^4^ cells per well of a 48-well plate coated with poly-L-lysine (Sigma-Aldrich), respectively.

For L1 function triggering, neurons were treated with 50 µg/mL L1 antibody 557 for 2 h at 37 °C. For immunoprecipitation, cells were seeded at a density of 3 × 10^6^ cells per well of a 6-well plate coated with poly-L-lysine, and cells from three wells were used per condition. For experiments with protease inhibitors, neurons were treated 2 h after seeding with 10 µM DAPT, GM6001 or LY2886721, or with 1 µM aprotinin for 24 h. Treatment of cerebellar and cortical neurons with 50 µg/mL tat-L1/KDET peptide was performed 30 min after seeding.

### 4.8. Transfection of Cortical Neurons and Determination of Neurite Outgrowth

After seeding, cortical neurons were maintained for 2 h before transfection with 400 ng plasmid containing L1 cDNA, 2 µL 10 µM siRNA, and 1 µL FuGENE transfection reagent per well. L1 antibody 557 (12.5 µg 557 per well) was added to the cultures 24 h after transfection, and 48 h after transfection, the cells were washed gently with pre-warmed culture medium, fixed in 2.5% glutaraldehyde, and stained with 1% toluidine blue and 1% methylene blue in 1% sodium tetraborate for 1 h at room temperature. Neurite outgrowth was analyzed by measuring the total length of neurites in an Axiovert microscope with the AxioVision 4.6 imaging system (Zeiss, Oberkochen, Germany).

### 4.9. Proximity Ligation Assay with Cerebellar and Cortical Neurons

Cultures were fixed for 15 min at room temperature in 4% formaldehyde, washed with PBS and subjected to proximity ligation assay using Duolink PLA products (Sigma-Aldrich) according to the manufacturer’s protocol (Sigma-Aldrich; Duolink PLA technology) with minor modifications. In brief, cells were blocked using Duolink Blocking solution supplemented with 0.5% Triton X-100 and incubated for 24 h at 4 °C with L1 antibody C-2 and MeCP2 antibody diluted 1:10 in Duolink Antibody Diluent. Cells were washed two times using Duolink Wash Buffer A and incubated with a mixture of secondary antibodies conjugated with oligonucleotides (Duolink Anti-Mouse PLA Probe MINUS and Duolink Anti-Rabbit PLA Probe PLUS). The proximity ligation reaction was then performed according to the manufacturer’s protocol using the Duolink In Situ Detection Reagent RED. Thereafter, the coverslips were incubated in 5 µg DAPI/mL PBS for 15 min, washed twice with PBS, and mounted in Immuno-Mount (Thermo Fisher Scientific). A total of 10 images per condition were taken using an Olympus F1000 confocal microscope (Olympus, Hamburg, Germany)and analyzed using ImageJ software (ImageJ, RRID:SCR_003070). The number of red puncta and the number of DAPI-stained nuclei were determined using ImageJ, and the number of red puncta per image was divided by the number of nuclei per image. The average values of red puncta per cell (=nuclei) were determined in 10 images per condition, and 200–500 cells per condition were analyzed.

### 4.10. qRT-PCR Analysis

For reverse transcription, oligoT18 primer and SuperScript^®^ II reverse transcriptase (Thermo Fisher Scientific) was used. qPCR was performed in triplicates using reverse transcribed mRNA, the 7900HT Fast Real-Time PCR System (Thermo Fisher Scientific), the qPCR kit SYBR^®^ Green I, ROX (Eurogentec, Cologne, Germany), and primers for the determination of the mRNA levels of BDNF (fw: GGC TGA CAC TTT TGA GCA CGT C; rev: CTC CAA AGG CAC TTG ACT GCT G), Wnt7a (fw: GGC TTC TCT TCG GTG GTA GC; rev: TGA AAC TGA CAC TCG TCC AGG), Hes5 (fw: AGT CCC AAG GAG AAA AAC CGA; rev: GCT GTG TTT CAG GTA GCT GAC), A2bp1 (fw: GCC CCT GAC ACA ATG GCT C; rev: GTC TGG CCG GTG TAC TCT G), Mef2c (fw: ATC CCG ATG CAG ACG ATT CAG; rev: AAC AGC ACA CAA TCT TTG CCT), and the reference gene actin (fw: TTC TGC ATC CTG TCA GCA ATG; rev: TCC TGT GGC ATC CAT GAA ACT). The SDS 2.4 software (Thermo Fisher Scientific) was used for analysis of the qPCR data. The mRNA levels of BDNF, Wnt7a, Hes5, A2bp1, and Mef2c relative to the mRNA levels of the reference gene actin were calculated. Data of relative gene expression (ΔCt values) were used for statistical analysis.

### 4.11. Statistical Analysis

Tests are indicated in the legends.

## Figures and Tables

**Figure 1 ijms-23-03554-f001:**
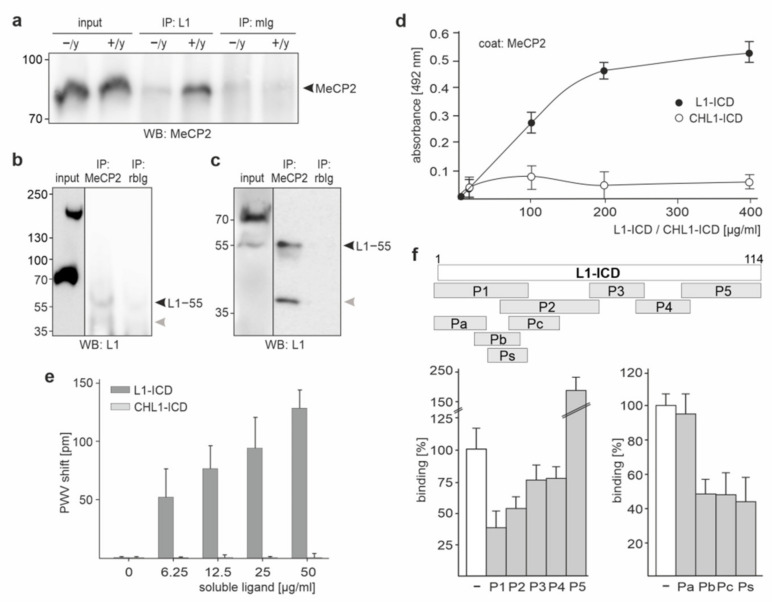
L1-ICD binds directly to MeCP2. Nuclear fractions were isolated from brains of wild-type (+/y) and L1-deficient (−/y) mice using protocol A (**a**,**b**) or B (**c**). These nuclear brain extracts were subjected to immunoprecipitation (IP) with L1 antibody C-2 (**a**), MeCP2 antibody (**b**,**c**), or non-immune control antibodies (mIg; rbIg) (**a**–**c**). Immunoprecipitates were then subjected to Western blot analysis with antibodies against MeCP2 (**a**) or L1 (**b**,**c**). Bands for MeCP2 (**a**) and L1−55 (**b**,**c**) are indicated by black arrowheads. Grey arrowheads indicate a band of ~40 kDa, which is recognized by the L1 antibody C-2. (**d**) Recombinant MeCP2 was substrate-coated and incubated with increasing concentrations of L1-ICD or CHL1-ICD. Binding was determined by ELISA using mouse L1 antibody C-2, goat CHL1 antibody, and horseradish peroxidase-conjugated secondary antibodies. Mean values ± SEM from three independent experiments carried out in triplicates are shown. (**e**) Recombinant MeCP2 was used as substrate-coat in a label-free binding assay and incubated with increasing concentrations of L1-ICD or CHL1-ICD. Binding was determined by measuring the peak wavelength (PWV) shift. Mean values + SD from three independent experiments carried out in triplicates are shown. (**f**) Recombinant MeCP2 was substrate-coated and incubated with L1-ICD in the absence (−) or presence of a 5-fold excess of L1 peptides P1, P2, P3, P4, or P5, which cover the entire L1-ICD sequence, or presence of a 5-fold excess of L1 peptides Pa, Pb, Pc, or Ps, which cover the N-terminal, membrane-proximal 43 amino acids of L1-ICD. The scheme indicates the positions of the peptides (grey bars) in L1-ICD (white bar) comprising 114 amino acids, which correspond to the amino acids 1147–1260 of mouse L1 (L1CAM_MOUSE, P11627). Binding was determined by ELISA using mouse L1 antibody C-2 and horseradish peroxidase-conjugated secondary antibodies. Mean values + SD from three independent experiments carried out in triplicates are shown for L1-ICD alone (white bars) and L1-ICD in the presence of peptide (grey bars).

**Figure 2 ijms-23-03554-f002:**
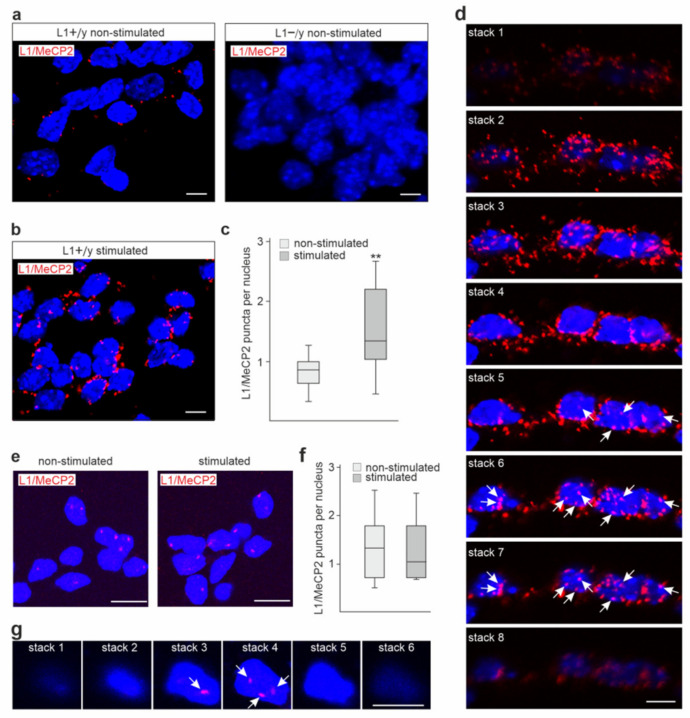
Interaction of L1 with MeCP2 in cultured cerebellar and cortical neurons. (**a**–**c**) Cultured cerebellar neurons from wild-type (L1+/y) (**a**,**b**,**d**) and L1-deficient (L1−/y) (**a**) mice were treated without (non-stimulated) or with (stimulated) function-triggering L1 antibody 557 and then subjected to proximity ligation assay with L1 and MeCP2 antibodies. (**e**,**g**) Cortical neurons from wild-type mice were treated without (non-stimulated) (**e**,**g**) or with (stimulated) (**e**) function-triggering L1 antibody 557. (**a**,**b**,**d**,**e**,**g**) Nuclei are stained with DAPI. Representative images of non-stimulated and stimulated L1+/y and L1−/y neurons are shown. (**d**,**g**) Images of z-axis stacks (stack 1 = top of the cell) are shown for representative stimulated cells stained with L1 and MeCP2 antibody. Arrows indicate purple puncta in nuclei. Scale bars: 10 µm. Red and purple puncta indicate close interaction of L1 with MeCP2 in cytoplasm and nucleus. (**c**,**f**) Box plots are shown for the average numbers of red puncta per nucleus from three independent experiments analyzing 200–500 cells per condition and experiment (** *p* < 0.01; Wilcoxon–Mann–Whitney U test).

**Figure 3 ijms-23-03554-f003:**
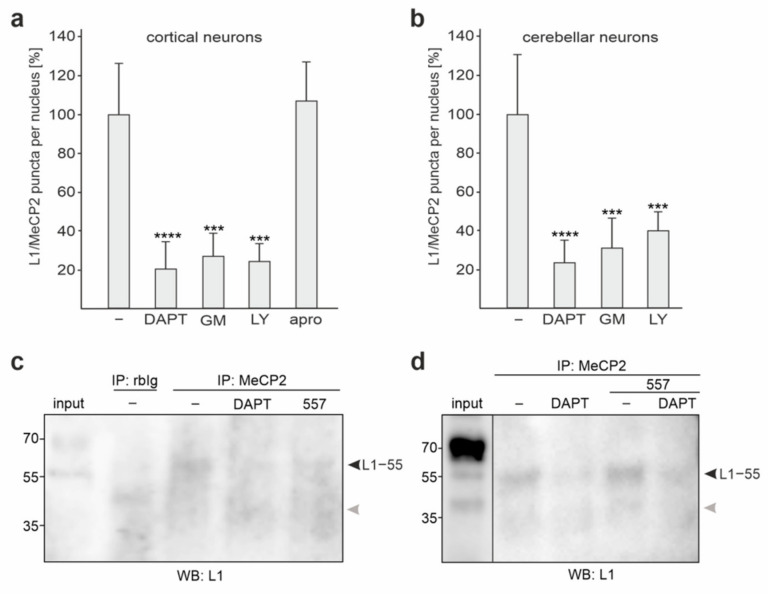
L1−55 is generated by α-, β-, and ɣ-secretases. Cultured cortical (**a**) and cerebellar (**b**) neurons were treated with vehicle (−) or with the protease inhibitors DAPT, GM60001 (GM), LY2886721 (LY), and aprotinin (apro) and then subjected to proximity ligation with L1 and MeCP2 antibodies. Mean values + SD from three independent experiments analyzing 200–500 cells per condition and experiment are shown for the average numbers of L1/MeCP2-positive red puncta per cell relative to the control (values of treatment with vehicle set to 100%) (*** *p* < 0.005, **** *p* < 0.001; one-way ANOVA with Bonferroni multiple comparison test). (**c**,**d**) Cultured cortical (**c**) and cerebellar (**d**) neurons were treated without or with DAPT and/or L1 antibody 557. Nuclear extracts were isolated from the treated neurons and used for immunoprecipitation (IP) with MeCP2 antibody and non-immune rabbit antibody (rbIg). The immunoprecipitates and the nuclear extracts (input) were subjected to Western blot analysis of the immunoprecipitates with L1 antibody C-2. Bands for L1−55 are indicated by black arrowheads. It is noted that a band of ~40 kDa was occasionally seen in in the L1 immunoprecipitates and nuclear extracts. This band is indicated by grey arrowheads.

**Figure 4 ijms-23-03554-f004:**
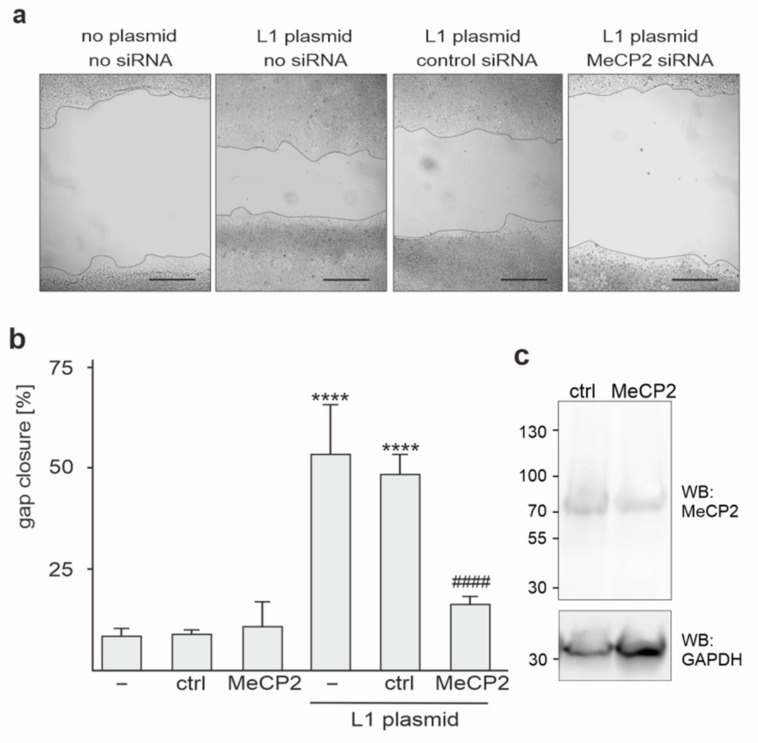
MeCP2 contributes to L1-dependent migration of HEK293 cells. (**a**–**c**) HEK293 cells were transfected without (−) or with control siRNA (ctrl) or MeCP2 siRNA alone, or with L1-encoding plasmid alone or together with control or MeCP2 siRNA. After 24 h transfected cells were subjected to a scratch wound assay, and the extent of gap closure was determined after further 24 h. (**a**) Representative images are shown for scratch wounds with dotted lines outlining the gaps. Scale bars: 100 µm. (**b**) Mean values + SD from three experiments carried out in duplicates are shown for gap closure, which was calculated as a ratio of the gap size 24 h after scratch injury and the gap size immediately after scratch injury (set to 100%) (**** *p* < 0.001 relative to mock-transfected cells, ^####^ *p* < 0.001 relative to the L1-expressing cells not co-transfected with siRNAs; one-way ANOVA with Dunn’s multiple comparison test). (**c**) Western blot analysis of lysates from cells co-transfected with L1 plasmid and control siRNA or MeCP2 siRNA.

**Figure 5 ijms-23-03554-f005:**
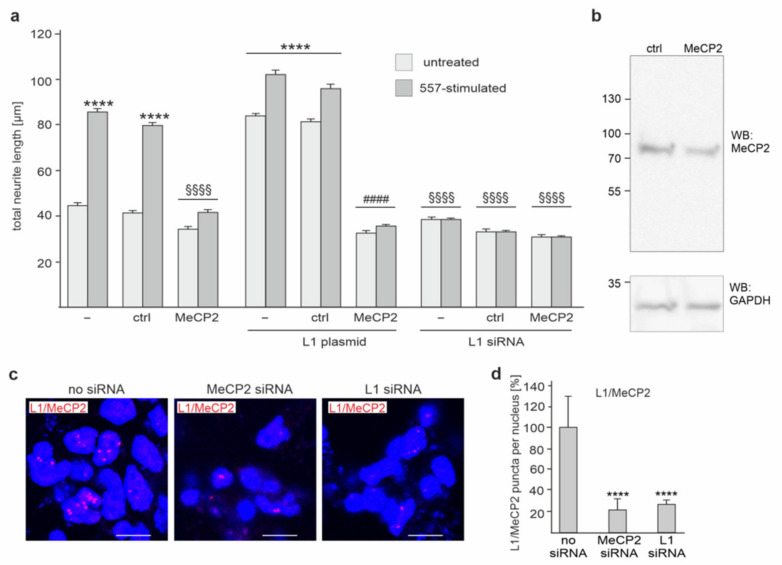
MeCP2 contributes to L1-dependent neurite outgrowth from cortical neurons. (**a**,**b**) Cortical neurons were transfected without (−) or with control siRNA (ctrl) or MeCP2 siRNA (MeCP2) alone, with L1-encoding plasmid (L1 plasmid) or L1 siRNA alone or together with control siRNA or MeCP2 siRNA. Neurons were then treated without or with L1 antibody 557. (**a**) Mean values + SEM from three independent experiments are shown for total neurite lengths (**** *p* < 0.001 relative to mock-transfected neurons, ^####^ *p* < 0.001 relative to neurons transfected with L1 plasmid alone, ^§§§§^ *p* < 0.001 relative to mock-transfected neurons treated with antibody 557; one-way ANOVA with Dunn’s multiple comparison test). (**b**) Western blot of lysates from cells transfected with control siRNA (ctrl) or MeCP2 siRNA (MeCP2). (**c**,**d**) Cortical neurons were transfected without (no siRNA) or with MeCP2 siRNA or L1 siRNA and then subjected to proximity ligation with L1 and MeCP2 antibodies. (**c**) Representative images of cells with L1/MeCP2-positive puncta (red) are shown (*n* = 3). Nuclei are stained in blue. Scale bars: 5 μm. (**d**) Mean values + SD from three experiments analyzing 200–500 cells per condition and experiment are shown for the average numbers of L1/MeCP2-positive red puncta per cell relative to the control (values after mock-transfection set to 100%) (**** *p* < 0.001; one-way ANOVA with Dunn’s multiple comparison test).

**Figure 6 ijms-23-03554-f006:**
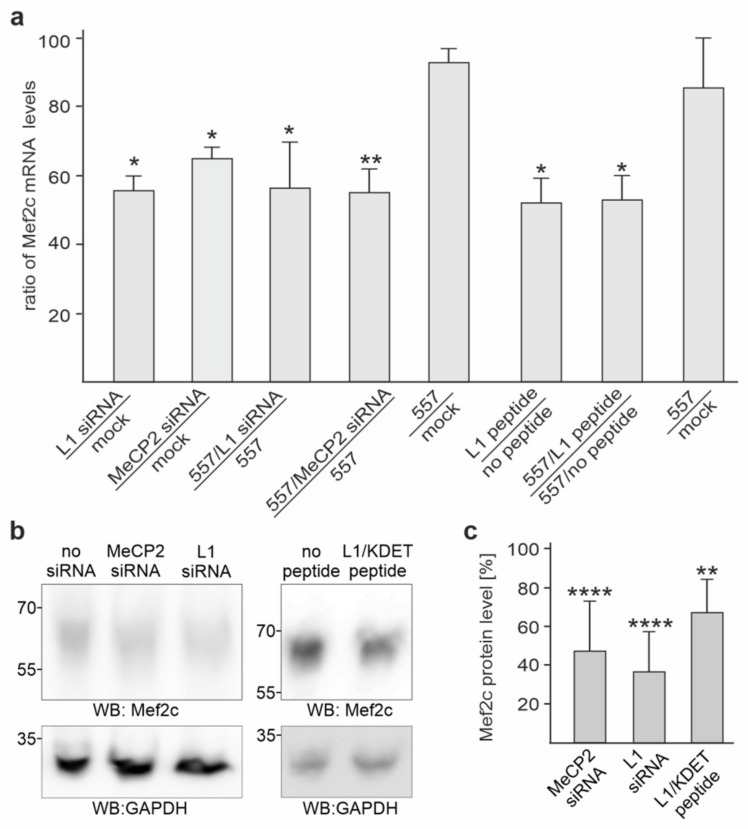
MeCP2 and L1 regulate Mef2c expression in cortical neurons. (**a**) Cortical neurons were mock-transfected or transfected with MeCP2 siRNA or L1 siRNA, or they were treated without or with the cell-penetrating tat-L1/KDET peptide. Neurons were then treated without or with L1 antibody 557 and subjected to RNA isolation and RT-qPCR. The Mef2c mRNA levels were normalized to the actin mRNA levels. Mean values + SD from three independent experiments with triplicates are shown for normalized Mef2c mRNA levels in neurons after transfection with siRNA or treatment with the tat-L1/KDET peptide relative to the levels in mock-transfected neurons (* *p* < 0.05; ** *p* < 0.01; one-way ANOVA with Dunn’s multiple comparison test). (**b**) Lysates from cortical neurons treated without or with MeCP2 siRNA, L1 siRNA, or tat-L1/KDET peptide were subjected to Western blot analysis with Mef2c antibody or GAPDH antibody to control loading. (**c**) Mean values + SD from four experiments are shown for Mef2c protein levels relative to control levels (values without treatment set to 100%) (** *p* < 0.01, **** *p* < 0.001; one-way ANOVA with Bonferroni’s multiple comparison test).

**Figure 7 ijms-23-03554-f007:**
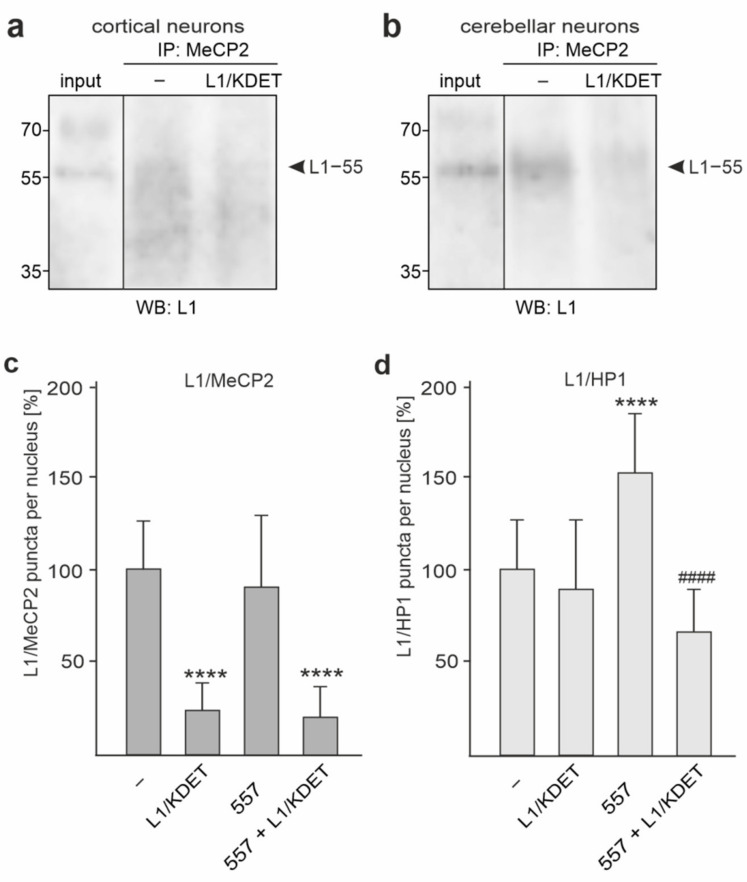
The MeCP2 and L1 interaction is disturbed by the tat-L1/KDET peptide. (**a**,**b**) Cultured cortical (**a**) and cerebellar (**b**) neurons were treated without (−) or with the cell-penetrating tat-L1/KDET peptide (L1/KDET). Nuclear extracts were isolated from the treated neurons and used for immunoprecipitation (IP) with MeCP2 antibody. The immunoprecipitates and the nuclear extracts (input) were subjected to Western blot analysis with L1 antibody C-2. Bands for L1−55 are indicated by black arrowheads. (**c**,**d**) Cortical neurons were treated without (−) or with cell-penetrating tat-L1/KDET peptide (L1/KDET), L1 antibody 557 alone (557), or L1 antibody 557 and tat-L1/KDET peptide (557 + L1/KDET). The transfected cells were then subjected to proximity ligation with an L1 and MeCP2 antibody (**c**). In parallel, proximity ligation with L1 and a mixture of HP1α, -β, and -γ antibodies was performed (**d**). Mean values + SD from three experiments analyzing 200–500 cells per condition and experiment are shown for the average numbers of L1/MeCP2-positive (dark grey bars) (**c**) or L1/HP1-positive (light grey bars) (**d**) red puncta per cell relative to the control (values without peptide set to 100%) (**** *p* < 0.001 relative to untreated neurons, ^####^ *p* < 0.001 relative to stimulated neurons; one-way ANOVA with Dunn’s multiple comparison test).

## Data Availability

Not applicable.

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
