# Peer review of "The Cell Adhesion Molecule L1 Interacts with Methyl CpG Binding Protein 2 via Its Intracellular Domain"

_ijms, 2022, doi:10.3390/ijms23073554_

Round 1

Reviewer 1 Report

This is a very well written manuscript that clearly explains the background and the context of the study.  The novely of the results are well described and the exeriments have the appropriate controls. The interaction has has been mapped by using recombinant proteins in direct binding assays. It is nice to see the validation of the co-immunoprecipitation experiments with PLA experiments in cultured cells followed by functional assays.

Minor comments:

  1. Annotations in the figures have a small font size and could therefore benefit from using high contrast colours. For example panel f in Figure 1, instead of white text on gray background use black text.
  2. Figure 2c. Change 'red spots' to 'L1:MeCP2 PLA puncta'
  3. Figure 3c, 7a, 7b. It is difficult to decipher the bands on these blots due to the high background.
  4. The authors state: 'Western blot analysis of cells transfected with control or MeCP2 siRNA and L1 plasmid showed reduced MeCP2 levels (18.7%; n=3) in L1-expressing cells after transfection with MeCP2 siRNA relative to L1-expressing cells transfected with control siRNA (Figure 4c)'.  Please add a statistical analysis to this experiment and similarly to the experiment in Figure 5b.
  5. For the PLA analysis please add information to the figure legends on how many cells per experiment were analysed. This is important information for the reader.

Author Response

Minor comments:

  1. Annotations in the figures have a small font size and could therefore benefit from using high contrast colours. For example panel f in Figure 1, instead of white text on gray background use black text.

Our response: We have enlarged the scheme and increased the font size of the labelling in Fig. 1f. In addition, we changed the background and font color for “L1-ICD”.

  1. Figure 2c. Change 'red spots' to 'L1:MeCP2 PLA puncta'

Our response: We have changed the label “number of red spots per nucleus” to “L1/MeCP2 puncta per nucleus” in Figure 2c and 2f. For consistency, we have also changed the label “spots per nuclei [%]”to “L1/MeCP2 puncta per nucleus [%]” in Figure 3a and 3b, the label “spots/cell [% of control]” to “L1/MeCP2 puncta per nucleus [%]” in Figure 5d, and the label “spots/cell [%]” to “L1/MeCP2 puncta per nucleus [%]” (left panel) or “L1/HP1 puncta per nucleus [%]” (right panel) in Figure 7d.

  1. Figure 3c, 7a, 7b. It is difficult to decipher the bands on these blots due to the high background.

Our response: We tried to improve the contrast and to reduce the background in Fig. 3c, 3d, 7a and 7b.

  1. The authors state: 'Western blot analysis of cells transfected with control or MeCP2 siRNA and L1 plasmid showed reduced MeCP2 levels (18.7%; n=3) in L1-expressing cells after transfection with MeCP2 siRNA relative to L1-expressing cells transfected with control siRNA (Figure 4c)'.  Please add a statistical analysis to this experiment and similarly to the experiment in Figure 5b.

Our response: We have now added the statistical analysis for the reduced MeCP2 levels in HEK293 cells and cortical neurons.

  1. For the PLA analysis please add information to the figure legends on how many cells per experiment were analysed. This is important information for the reader.

Our response: We have now added the information to the figure legend that 200-500 cells per condition were counted for PLA analysis.

Reviewer 2 Report

In the manuscript “The cell adhesion molecule L1 interacts with methyl CpG binding protein 2 via its intracellular domain” Loers and coauthors have showed the direct binding of MeCP2 protein to intracellular domain of Cell Adhesion L1 molecule and have suggested the binding epitope within L1. Using various in vitro approaches authors showed that the MeCP2-L1 interaction regulate cellular migration and neurite outgrowth, well-known functions attributed to L1.

The manuscripts has a good structure and well-written. Conclusions are supported by results. Multiple controls and original Western blot images were provided.  

Minor comments:

  • In the result section 2.1, authors indicated peptides following the amino acid fragments : “Peptides comprising L1-ICD amino acids 1-35, 14-29, 19-32, 23-57 and 25-43 reduced the binding of L1-ICD to MeCP2, whereas peptides comprising amino acids 1-17, 55-73 and 70-89 did not affect the binding (Figure 1f)”. However, both in the Fig 1f and in the Method section, the peptides were indicated with P1, P2, etc. Could be easier for readers to add here also the ”P” name of the peptides after the amino acid numbers, like “…amino acids 1-35 (P1) …”. What peptide name/number was allocated for 87-114 fragment “A peptide comprising amino acids 87-114  …”?
  • Fig. 1f, legend: “Recombinant MeCP2 was substrate-coated and incubated with L1-ICD in the absence (-) or presence of a 5-fold excess of L1 peptides P1, P2, P3, P4 or P5, which cover the entire L1-ICD sequence, of a 5-fold excess of L1 peptides Pa, Pb, Pc, or Ps, which cover the N-terminal,…” do authors mean “or” instead of “of”? The peptide labelling on the Fig. 1f scheme is very small and “L1-ICD” name is difficult to read – it would be nice to have this drawing more clear.
  • It would be nice to indicate in the Method section the protein ID number for the L1 sequence used for peptide design. Does L1 sequence conservative across species?

Author Response

Minor comments:

  • In the result section 2.1, authors indicated peptides following the amino acid fragments : “Peptides comprising L1-ICD amino acids 1-35, 14-29, 19-32, 23-57 and 25-43 reduced the binding of L1-ICD to MeCP2, whereas peptides comprising amino acids 1-17, 55-73 and 70-89 did not affect the binding (Figure 1f)”. However, both in the Fig 1f and in the Method section, the peptides were indicated with P1, P2, etc. Could be easier for readers to add here also the ”P” name of the peptides after the amino acid numbers, like “…amino acids 1-35 (P1) …”. What peptide name/number was allocated for 87-114 fragment “A peptide comprising amino acids 87-114  …”?

Our response: We have now added the peptide name after the amino acid numbers as proposed by the reviewer. We changed “A peptide comprising amino acids 87-114  …” to “The peptide comprising amino acids 87-114 (P5)…”.

  • Fig. 1f, legend: “Recombinant MeCP2 was substrate-coated and incubated with L1-ICD in the absence (-) or presence of a 5-fold excess of L1 peptides P1, P2, P3, P4 or P5, which cover the entire L1-ICD sequence, of a 5-fold excess of L1 peptides Pa, Pb, Pc, or Ps, which cover the N-terminal,…” do authors mean “or” instead of “of”? The peptide labelling on the Fig. 1f scheme is very small and “L1-ICD” name is difficult to read – it would be nice to have this drawing more clear.

Our response: We changed the sentence to “…in the absence (-) or presence of a 5-fold excess of L1 peptides P1, P2, P3, P4 or P5, which cover the entire L1-ICD sequence, or presence of a 5-fold excess of L1 peptides Pa, Pb, Pc, or Ps, which cover the N-terminal, membrane-proximal 43 amino acids of L1-ICD.

We have enlarged the scheme and changed the font color and background color for “L1-ICD” in Fig. 1f.

  • It would be nice to indicate in the Method section the protein ID number for the L1 sequence used for peptide design. Does L1 sequence conservative across species?

Our response: We had already indicated the ID number for the L1 sequence in the Methods section. We have now also mentioned the ID number in the legend: “The scheme indicates the positions of the peptides in L1-ICD comprising 114 amino acids which correspond to the amino acids 1147-1260 of mouse L1 (L1CAM_MOUSE, P11627)”.